

# Flood frequency analysis using mean daily flows vs. instantaneous peak flows

Anne Bartens[1], Uwe Haberlandt[1]

[1]Institute of Hydrology and Water Resources Management, Leibniz University of Hannover, Germany

*Correspondence to*: Anne Bartens (fangmann@iww.uni-hannover.de)

**Abstract.** In many cases flood frequency analysis needs to be carried out on mean daily flow (MDF) series without any available information on the instantaneous peak flow (IPF). We analyze the error of using MDFs instead of IPFs for flood quantile estimation on a German dataset and assess spatial patterns and factors that influence the deviation of MDF floods from their IPF counterparts. The main dependence could be found for catchment area but also gauge elevation appeared to

have some influence. Based on the findings we propose simple linear models to correct both MDF flood peaks of individual flood events and overall MDF flood statistics. Key predictor in the models is the event-based ratio of flood peak and flood volume obtained directly from the daily flow records. This correction approach requires a minimum of data input, is easily applied, valid for the entire study area and successfully estimates IPF peaks and flood statistics. The models perform particularly well in smaller catchments, where other IPF estimation methods fall short. Still, the limit of the approach is reached

for catchment sizes below 100 km², where the hydrograph information from the daily series is no longer capable of approximating instantaneous flood dynamics.

## 1 Introduction

Common flood frequency analysis (FFA) is based on samples of maximum flows. The dimensions and variability of these maxima pose the baseline for the choice of probability distribution, the estimation of its parameters and eventually the

deduction of flood quantiles as design criteria. For FFA to be as accurate as possible, it is important to have a large number of peak flows measured with high precision, so that flood magnitude and dynamics are well assessable.

However, embracing the true dimension of a peak requires continuous measurement of the flow on a high temporal resolution. Such data is rarely available and oftentimes FFA needs to be carried out on average daily flow records instead. The daily averaging naturally flattens the flood peak and the true maximum becomes unknowable. The degree of this smoothing, i.e. the

difference between the true instantaneous peak flow (IPF) and the maximum mean daily flow (MDF) depends on the response time of a system, which is controlled by a multitude of factors. The average relationship between MDF and IPF peaks at a site depends greatly on its basin area (Fuller, 1914) and characteristics related to topography, like altitude, relief and channel slope (Canuti and Moisello, 1982). The internal variability of the MDF-IPF ratio within a site's flow record is largely determined by the type of meteorological input causing the individual flood events (Viglione and Blöschl, 2009; Gaál et al. 2013). A variety



of studies make use of the dependencies named above in order to estimate IPFs from MDFs, including Taguas et al. (2008), Muñoz et al. (2012) and Ding et al. (2015).

Other IPF estimation methods aim at using the bare minimum of available data, i.e. solely the available daily flow record. In these cases, usually the shape of hydrographs are used to estimate the instantaneous peaks of events (e.g. Langbein, 1944; Ellis and Gray, 1966). Several approaches make use of the maximum daily flow and the discharge of the previous and successive

day. Chen et al. (2017) compare two of these methods, namely those of Sangal (1983) and Fill and Steiner (2003). They also propose their own method based on the rising and falling slopes of the event hydrograph, estimated from the three consecutive days around the peak. They found that their slope-based method and Fill and Steiner's method perform well and are probably applicable under a wide range of climates. However, both methods' performances decrease with decreasing catchment size and work best for areas larger than 500 km².

There naturally exist more complex means to correct the divergence between MDFs and IPFs. This includes disaggregation of the daily flow series to a finer scale, as done by e.g. Stedinger and Vogel (1984), Tarboton et al. (1998), Kumar et al. (2000), Tan et al. (2007) and Acharya and Ryu (2014). Also, hydrological modelling may be applied for IPF estimation, e.g. in combination with high-resolution disaggregated rainfall (Ding et al., 2016) and regionalized model parameters (Ding and Haberlandt, 2017). These methods, however, require a variety of computational steps and/or data sources.

This study aims at analyzing the differences between IPF and MDF with focus on flood frequency. The errors in mean maximum flows, distribution parameters and flood quantiles are assessed and analyzed for spatial patterns. Based on the findings, a method is proposed that facilitates IPF estimation using a combination of daily event hydrographs and functional dependencies with geomorphic catchment descriptors, while keeping the data input to a minimum. Key predictor in this approach is the ratio of direct event peak runoff and direct event volume. This ratio is expected to effectually describe the

shape of a flood event, which in turn gives an idea about the expected instantaneous peak: the larger the daily peak and the smaller the event volume, the larger the expected difference between IPF and MDF and vice versa. We assume that the peak-volume ratio (p/V) holds important information on the general behavior of flood events (Tan et al., 2006; Gaál et al. 2015; Fischer, 2018), and thus the expected magnitude of the IPF. The p/V of individual events can describe the internal variability at a site by reflecting different types of floods caused by different rainfall and/or snowmelt inputs. At the same time the p/V

accounts for the variability between sites caused by local flood generating processes governed by general physiographic conditions. Accordingly, the proposed method is tested for IPF estimation for individual events, which are then used for FFA, and for direct correction of site-specific distribution parameters and flood quantiles.

## 2 Methods

### 2.1 Analysis and estimation of IPF peaks

In a first step, the general differences between IPF and MDF statistics are analyzed. An error is computed as percentage deviation of the MDF statistic $MDF_{stat}$ from the IPF statistic $IPF_{stat}$



$$\text{Error} = \frac{\text{MDF}_{\text{stat}} - \text{IPF}_{\text{stat}}}{\text{IPF}_{\text{stat}}} * 100 \ \%. \tag{1}$$

This error is computed at each station for any desired quantity *stat*, like the mean annual maximum flow (MHQ), L-moments,
distribution parameters and flood quantiles.

In order to improve the IPF estimation by MDF, several correction methods are applied, which make use of the peak-volume
ratio. This ratio is computed for events in the average daily time series after baseflow separation using the direct peak flow
$Q_{dir}$ and the direct flood volume $Vol_{dir}$

$$\text{p/V} \left[\tfrac{1}{d}\right] = \frac{Q_{\text{dir}} \ [\text{m}^3\text{d}^{-1}]}{\text{Vol}_{\text{dir}} \ [\text{m}^3]}. \tag{2}$$

The first IPF estimation method aims at correcting individual events. For calibration, all events are identified that contain a
monthly maximum instantaneous peak. For these events the daily and instantaneous peaks, as well as the daily p/Vs are
computed. Then a linear regression model of the following form is fitted

$$\text{IPF}_{\text{event}} = \text{MDF}_{\text{event}} * (a + b_1 * \text{p/V}_{\text{event}} + b_2 * \text{CD}_1 + \cdots + b_{n+1} * \text{CD}_n), \tag{3}$$

where *CD* denotes additional catchment descriptors that may be included in the models. The combination of hydrograph shape
and catchment characteristics as predictors is expected to better reproduce both the at-site and between-site variability in the
IPF-MDF relationship and yield a more universal model.

The event correction method will be compared with the slope correction method developed by Chen et al. (2017). This method
estimates an instantaneous event peak flow based on the slopes of the daily peak $Q_{peak}$ to its preceding and following daily
flows $Q_{pre}$ and $Q_{suc}$. The IPF is thus estimated as

$$\text{IPF}_{\text{event}} = Q_{\text{peak}} + \frac{(Q_{\text{peak}} - Q_{\text{pre}}) * (Q_{\text{peak}} - Q_{\text{suc}})}{2 * Q_{\text{peak}} - Q_{\text{pre}} - Q_{\text{suc}}}. \tag{4}$$

For validation, both methods are applied to estimate IPFs for all separated events in the daily flow series. The maximum flows
needed for FFA are then taken from this corrected event series rather than from corrected maximum daily peaks. This procedure
is assumed to be more accurate, since maxima in IPF and MDF do not necessarily overlap. More precisely, events with
maximum instantaneous peaks can have rather inconsiderable daily peaks in some instances. Correcting only the maximum
MDFs would lead to underestimation of the IPFs in these cases.

Since the p/V method in equation (3) can only be calibrated on events for which monthly maxima exist but is eventually
applied to all events, including very small and potentially improperly separated ones, unrealistic IPF estimates may be created
that adversely affect the subsequent FFA. In order to avoid this problem and to be able to estimate IPF statistics directly from




daily records, a second type of IPF estimation methods is analyzed. These involve the estimation of flood statistics, i.e. mean
annual and seasonal maximum flows, sample L-moments, estimated distribution parameters and derived flood quantiles based
on averaged p/Vs. These averages are obtained from all annual maximum MDF events at each station. These major events are
expected to be properly separated by the algorithm and to be most influential in FFA. Although the maximum MDF events
may not necessarily be identical to the maximum IPF events, as discussed before, this approach appears to be the most sensible
here. The model set up is analogous to the event correction approach

$$\text{IPF}_{\text{stat}} = \text{MDF}_{\text{stat}} * (a + b_1 * \text{p/V}_{\text{mean}} + b_2 * \text{CD}_1 + \cdots + b_{n+1} * \text{CD}_n).$$      (5)

The model is expected to represent the average conditions that determine the average deviation of MDF from IPF estimates.
The $p/V_{mean}$ in itself is expected to be a good predictor that reflects local conditions like spatial scale, climate, geology and
other external factors that control flow variability obtainable from daily flow records. The additional inclusion of catchment
descriptors is tested case by case and may contribute to the reproduction of spatial variability.

## 2.2 Event separation

For separation of the flood events, the initial steps of the procedure used by Tarasova et al. (2018) are carried out, which has
proven effective and convenient for their German dataset. For the initial step of baseflow separation they selected the simple
nonparametric algorithm by the Institute of Hydrology (1980). This method is applied here with the same settings, i.e. 5-day
non-overlapping blocks are used to find minima that are identified as turning points if they are more than 1.1 times smaller
than their neighboring minima. The baseflow is then derived by linear interpolation between the turning points. Discharge
peaks are subsequently determined from the flow series and for every peak the start and end of the belonging flow event is
defined by the nearest surrounding turning points. In order to prevent false identification of events due to natural variability,
events are discarded if their peak discharge is not at least 10% larger than the baseflow.

The final step of re-defining events with multiple peaks is not carried out, as it requires rainfall and snowmelt information,
which are not available in our case. It is assumed that the majority of events, especially the larger ones relevant for FFA, are
separated correctly.

## 2.3 Distribution fitting and flood quantile estimation

For extrapolation of the time series and estimation of floods with specific return periods, distributions were fitted to the annual
and seasonal samples of both IPF and MDF. This enables the direct comparison of both the higher flood quantiles and of the
estimated distribution parameters. Here, the General Extreme Value distribution (GEV) of the following form was used for all
samples

$$F(x) = e^{-\exp\left(\frac{1}{k} * \log\left(1 - k * \frac{x - \xi}{\alpha}\right)\right)}$$      (6)





with location parameter $\xi$, scale parameter $\alpha$ and shape parameter $k$. The parameters were estimated using sample L-moments. The goodness of fit of the distributions was determined with the Cramer-von-Mises test.

Additionally, for seasonal considerations, mixed models were applied, which combine two or more GEV distributions fitted to different subsamples of the data, like summer and winter floods. A simple maximum mixing approach is used to combine the individual distributions:

$$F_{\mathrm{mix}}(x) = \prod_{i=1}^{n} F_i(x). \tag{7}$$

This approach allows the combined estimation of flood quantiles from multiple underlying distributions and thus the
assessment of errors in seasonal FFA. The approach is described in detail in Fischer et al. 2016. Other than in their study, we do not censor our data with thresholds, i.e. for matters of simplicity we assume that every seasonal maximum is indeed a flood event.

### 2.4 Uncertainty

Since both distribution fitting and IPF estimation via linear models are approximations and not fully accurate, we eventually
assess the overall level of uncertainty in the final IPF flood quantile estimates. This is done using simple bootstrapping procedures. In a first step, the series of annual maxima from both daily and monthly maximum data are analogously resampled 1000 times with replacement. For each resampling the desired flood quantiles are estimated using L-moments. The range of these estimates provides the baseline level of uncertainty due to distribution fitting.

In a second step, linear regression models are fitted to each pairing of estimated IPF and MDF flood quantiles over all stations
in the study area. In order to assess the uncertainty of the fitted models, another resampling is carried out, this time shuffling the set of considered stations, again 1000 times with replacement. For each station this procedure yields 1000 estimates of paired flood quantiles from both the IPF and MDF series (IPF-bs and MDF-bs), 1000 full-model quantile estimates resulting from the original p/V model fitted to each permutation (p/V-full), and 1000*1000 quantile estimates resulting from permutation of the p/V model for all IPF and MDF transpositions (p/V-bs-bs).

In order to assess the overall level of uncertainty, several indices will be assessed at the individual stations. The first one is the relative width of the 95% confidence intervals (CI) calculated for all aforementioned bootstrap sample estimates of the desired flood quantile

$$CI_{\mathrm{bs}} = \frac{x_{\mathrm{bs};0.975} - x_{\mathrm{bs};0.025}}{x_{\mathrm{bs};0.5}}, \tag{8}$$

where $x_{\mathrm{bs};0.025}$ and $x_{\mathrm{bs};0.975}$ are the 2.5% and 97.5% quantile and $x_{\mathrm{bs};0.5}$ the median of the respective sample. The second one is
the deviation of the individual MDF and p/V-model bootstrap samples from the IPF sample, which allows the assessment of error distributions


$$\text{error}_{\text{bs}} = \frac{x_{\text{bs}} - \text{IPF}_{\text{bs}}}{\text{IPF}_{\text{bs}}} * 100\%. \tag{9}$$

From the resulting error vector, a variety of statistics can be computed for comparison. Finally, the agreement of the 95% confidence intervals of the MDF and p/V-model samples with the IPF confidence bands are determined as percentage overlap:

$$\text{overlap} = \frac{\min(x_{\text{bs};0.975}, \text{IPF}_{\text{bs};0.975}) - \max(x_{\text{bs};0.025}, \text{IPF}_{\text{bs};0.025})}{\max(x_{\text{bs};0.975}, \text{IPF}_{\text{bs};0.975}) - \min(x_{\text{bs};0.025}, \text{IPF}_{\text{bs};0.025})} * 100\%. \tag{10}$$

## 3 Study area and data

This study uses data from 653 discharge gauges distributed over Germany. For the analyses, average daily flow and maximum monthly flow are required. The selected stations represent the datasets of the federal agencies, who provide online access to both parameters (Lower Saxony, Saxony-Anhalt, Saxony, Bavaria and Baden-Württemberg).

Germany poses a transition zone from an oceanic climate in the northwest to a humid continental climate in the southeast. The northwestern parts are influenced by wet air and have mild winters, while the more southeastern parts are drier and exhibit larger temperature ranges. The average temperature for the entire country is 8.9 °C, the monthly averages ranging between 0.4°C in January and 18°C in July (reference period 1981-2010; DWD). The average precipitation is 819 mm, where amounts generally decrease in west-east direction and in strong dependence on topography. Annual rainfall sums are generally highest

over the Alps at the very Southern border and the various secondary mountain ranges. The flat continental east is driest. Temporally, the summer months are wettest with rainfall often occurring in convective events. Snowfall occurs between October and April, where amount and depth of snow cover increase with decreasing oceanic influence and increasing altitude. Even though not the entire area of Germany is covered by the available data, the selected gauges provide a cross section through the climatically and topographically distinct regions, from the flat oceanic northwest to the mountainous continental

southeast.





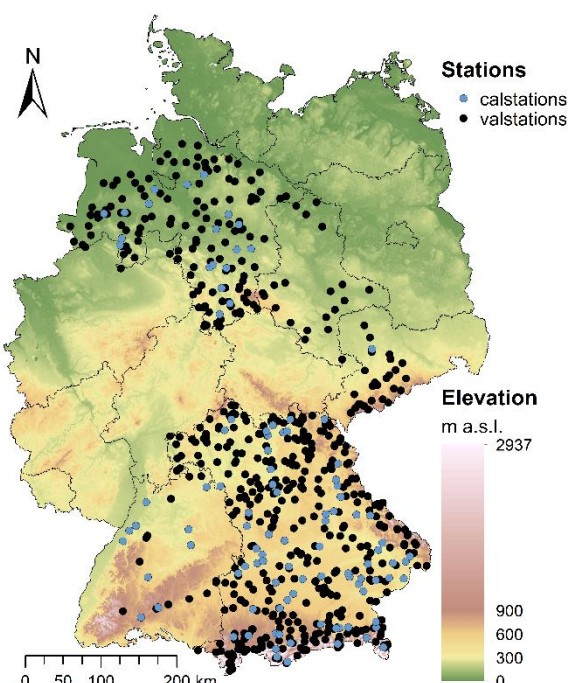

**Figure 1: Location of the 653 gauges used for analysis. The 103 stations used for model calibration are marked in blue. Digital elevation data by Jarvis et al. (2008).**

The lengths of the discharge records vary substantially from 11 to 183 years with a mean of 48.4 years. For the general assessment of differences in IPF and MDF floods and final model validation, all 653 stations with their variable record lengths are considered. For assessment of flood frequency criteria only those stations with at least 30 years of observations were used (490). Model fitting was carried out on a subset of 103 gauges, whose discharge series were thoroughly checked. Also, their records were cropped to a common period from 1979 to 2012, in order to eliminate potential non-stationary effects.

For the 103 stations used for calibration a catalogue of catchment descriptors is available. For the remaining stations only rudimentary information was obtained, i.e. catchment size, geographical position and altitude of the gauges.

## 4 Results and discussion

### 4.1 Comparison of MDF and IPF peaks

In theory, the relative deviation between MDF and IPF peaks depends greatly on catchment size. Small catchments without appreciable buffering capacity react fast to even small rainfall, leading to short and steep flood waves that are hardly reproduced on coarsely averaged time scales. Factors like steep slopes, impermeable underground and short but intense rainfall contribute to the "burstiness" of storm events and make these even less representable through daily flow records.

The effect of the catchment size is clearly visible in the data set. Fig. 2 demonstrates by means of the mean annual maximum flow that the larger the area, the smaller the deviation between MDF and IPF. Also, errors appear to be especially large in



higher altitudes. Generally, the error seems to increase in north-south direction, which could be a secondary effect of both

increasing altitude and decreasing catchment size.

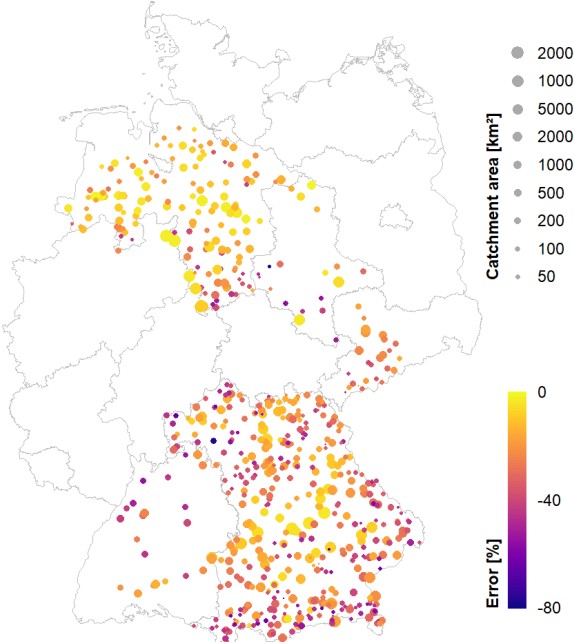

**Figure 2: Spatial distribution of the MDF error in the MHQ.**

When assessing the differences between average daily and instantaneous peaks, it is also meaningful to take a closer look at different types of floods. For our German dataset the two most opposite types are a) flood events induced by short intense

rainfall, especially convective events, and b) extended flood events with significant volume, as caused by snowmelt and/or stratiform rain. Presumably, the latter flood type is much better represented by average daily flow than the former. In order to roughly distinguish between the two types, the flow records are divided into summer (May - October) and winter (November - April) half years.





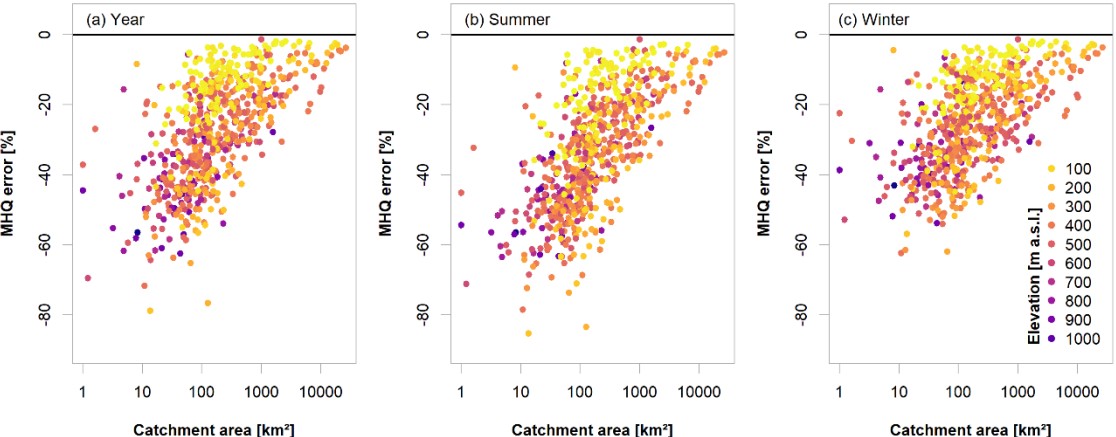

**Figure 3: Error in the MHQ in relation to catchment size and elevation for the entire year (a), summer (b) and winter (c).**


In Fig. 3 the error in the MHQ is shown for the entire year and the summer and winter half year. The relationship with catchment area is clearly visible in all three cases. Also, the effect of the elevation becomes obvious, namely in the lowest elevations (yellow points, below 100 m) showing very small errors, even for small catchment sizes down to approximately 100 km². This is the clearest stratification in the error due to elevation; the errors at higher altitudes appear less distinguishable.

There is, however, a clear distinction between summer and winter. As expected, the error is overall smaller in the winter months, where snowmelt and stratiform events prevail, while the convective events in summer are poorly reproduced by MDF. The error in the annual peaks is a mixture of the two seasons. Which season contributes mainly to the annual peaks depends on the individual flood regimes. At 69.1% of the considered gauges the winter floods exceed their summer counterparts on average, the remaining 30.9% are dominated by summer floods. These seasonality statistics are established on basis of the IPF.

When considering MDF instead, only 23.2% of the gauges are identified as having maximum peaks in summer. This indicates that the average daily flow smooths significant peaks to a point where they are no longer relevant for the overall flood behavior. Fig. 4 a shows the percentage of annual maxima at each gauge that are attributed to the wrong season. Negative values are falsely attributed to summer, positive values to winter. It is obvious that with decreasing catchment size an increasing number of annual maxima are falsely identified in the winter half year, while the actual instantaneous maxima occur in summer. Apart

from not being able to properly identify flood magnitudes when using daily flow series, this is a serious issue for classification of flood regimes, identification of dominating flood types and application of heterogeneous flood frequency analysis when daily data is the only available option.

Another general issue highlighted by this analysis, independent of seasonality, is the asynchronous occurrence of IPFs and MDFs. Instantaneous maxima are not always identifiable in the daily flow series, i.e. the maxima obtained from the daily series

are inevitably found in other places. In general, the smaller the catchment, the smaller the temporal overlap between instantaneous and daily peaks, as seen in Fig. 4 b. This problem needs to be kept in mind when attempting to estimate





instantaneous peaks from daily peaks, since the two may belong to significantly different events and thus to different populations.

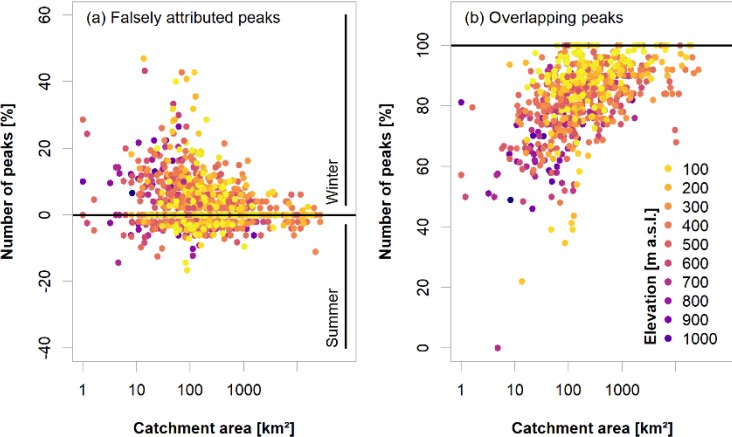

**Figure 4: Percentage of peaks falsely attributed by MDF to the winter or summer half-year (a) and percentage of peaks in MDF and IPF overlapping in time (with a 5-day buffer; b).**

### 4.2 Estimation of mean annual IPF

For both correction of the individual events and of the MHQ, linear models appeared appropriate. The fitted models are listed in Table 1. For all models, the p/V is able to explain the majority of the variance in the IPF-MDF relationship and they may well be applied with the p/V as sole regressor. However, including the gauge elevation in some models led to minor improvements of the performances, especially for the two seasons. Gauge location and catchment size, on the other hand, did not prove relevant.

It turned out that when correcting the peaks of individual events, the performance could not be significantly improved by considering the seasons separately. For correction of the MHQ, however, individual seasonal models did prove meaningful. This observation indicates that the relationship between average IPF, MDF and p/V differs between seasons. This is attributed to the inability of the daily series to identify instantaneous events, which is more severe in summer than in winter. The individually fitted models tend to correct for this deficiency.

**Table 1: Linear models fitted for correction of individual events and the MHQ.**

| Type | | Model |
|---|---|---|
| Events | | $MDF * (0.94 - 0.49 * p/V_{event} - 0.000078 * elevation)$ |
| MHQ | Year | $MHQ_{MDF} * (1.11 - 1.27 * p/V_{mean})$ |
| | Summer | $MHQ_{MDF} * (1.34 - 1.54 * p/V_{mean} - 0.00011 * elevation)$ |
| | Winter | $MHQ_{MDF} * (1.34 - 1.63 * p/V_{mean} - 0.00019 * elevation)$ |

Fig. 5 shows the change in mean absolute error in the annual MHQ after correction with the different methods in relation to catchment size and elevation. The slope method (a) applied to the individual events yields a rather constant reduction of the





error independent of catchment size. However, there are several outliers produced by the method, which can be attributed to improper separation of smaller events. Applying the slope method only to the annual maximum MDF events, as done in Fig.

5 b shows a much smoother and more constant error reduction. The two methods using the p/V ratio (Fig. 5 c and d) yield a much larger improvement for the smaller catchments where the error is generally larger than in the bigger catchments. However, both methods simultaneously lead to an increase of the error in several cases. This deterioration appears to affect those stations that have been highlighted earlier, namely the ones with the lowest elevations in the data set. Here the error is significantly lower than the p/Vs suggest. Again, the average p/V method appears more robust than the event-correction and

leads to larger improvements in smaller catchments.

It should be noted that working with large data and automatic event separation without manual post-correction leads to problems that could potentially be avoided when considering individual time series more carefully. Several events are identified as too long or too short (or not at all), so their volumes are over- or understated, respectively. This results in false p/Vs and in some cases to severe over- or underestimation of the peak. The weight of such events is assumed to be significantly

lower when correcting flood statistics based on average p/Vs, which could be a particular reason for the latter method being favored here. In addition, the overall performance can only be assessed for events that contain the monthly maximum flow, i.e. primarily larger events. How the event correction performs for minor events cannot be analyzed here.

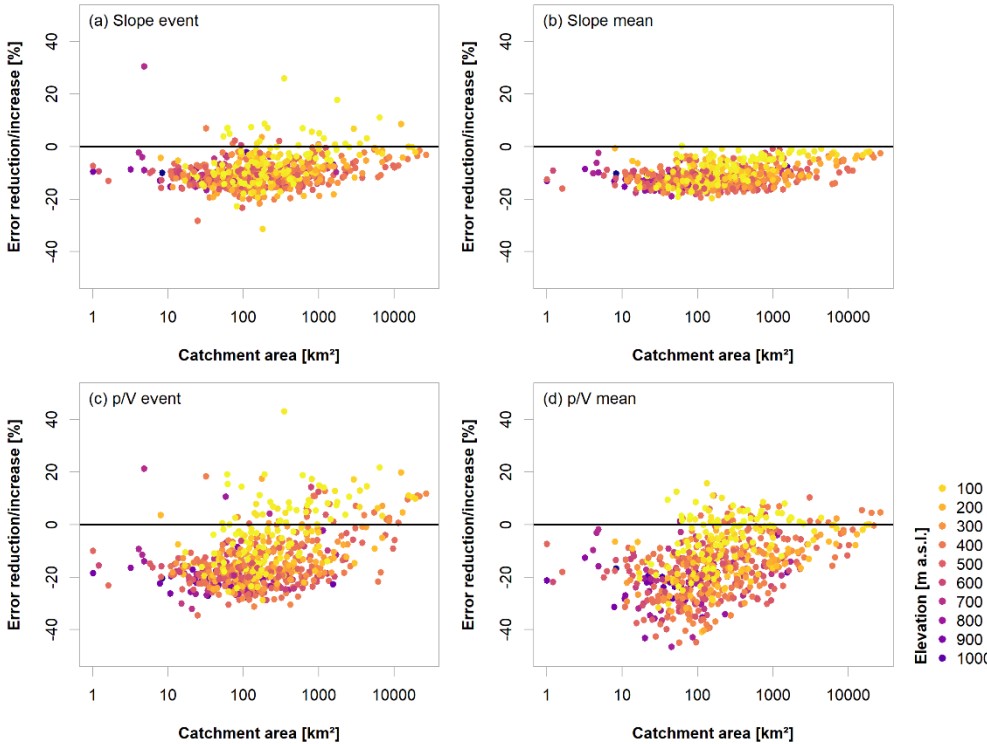

**Figure 5: Error reduction (negative values) / increase (positive values) in the mean maximum flow for different IPF estimation**
**methods when compared to MDF.**


Fig. 6 summarizes the overall model performances for the mean annual/seasonal maximum flow at all 653 stations and compares the individual methods to the error in using MDF directly. It is obvious that all methods give significantly better IPF estimates than the mere MDFs. The slope correction has quite a large bias, which is, as seen above, not only disadvantageous. Still, the overall error is smaller for the p/V methods, with fewer outliers produced by the average p/V approach.

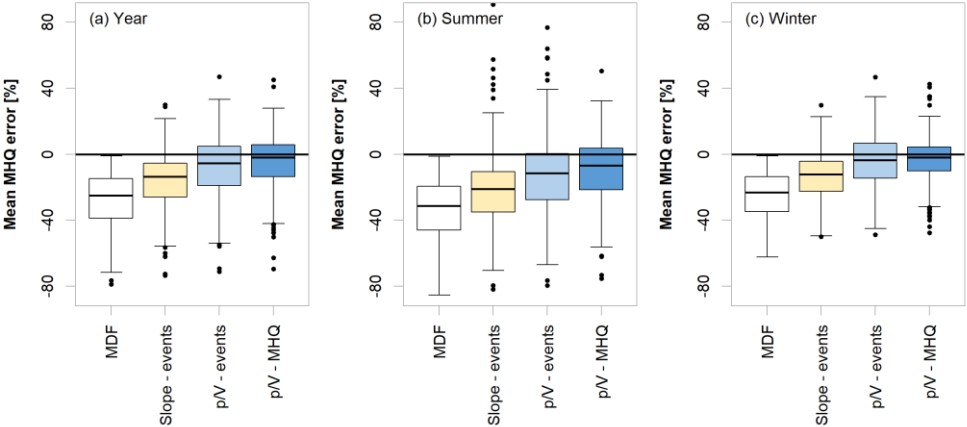


**Figure 6: Comparison of performances of different IPF estimation methods for the entire year (a), summer (b) and winter(c).**

Table 2 summarizes the normalized root mean square error (NRMSE) and the percentage bias (PBIAS) for all model variants. The values in parentheses indicate the performance criteria for gauges with catchment areas below 500 km². Here, the advantage of both p/V-approaches over the slope method become apparent, as their errors are significantly smaller.


**Table 2: Performances of different IPF estimation methods in terms of NRMSE and percentage bias. The values in parentheses show the performances for catchment sizes below 500 km².**

|  | Year | | Summer | | Winter | |
|---|---|---|---|---|---|---|
|  | NRMSE [%] | PBIAS [%] | NRMSE [%] | PBIAS [%] | NRMSE [%] | PBIAS [%] |
| MDF | 17.0 *(48.0)* | -18.0 *(-32.5)* | 18.1 *(49.1)* | -20.6 *(-38.2)* | 14.9 *(44.3)* | -16.4 *(-28.8)* |
| Slope-events | 9.2 *(32.4)* | -8.6 *(-20.5)* | 8.9 *(34.6)* | -11.6 *(-26.7)* | 10.3 *(30.3)* | -7.8 *(-17.3)* |
| p/V-events | 12.1 *(24.6)* | 2.0 *(-12.3)* | 11.5 *(26.7)* | -0.1 *(-18.0)* | 13.5 *(23.1)* | 2.9 *( -9.0)* |
| p/V-MHQ | 7.6 *(21.3)* | 0.7 *( -8.1)* | 8.2 *(23.4)* | -0.4 *(-12.3)* | 8.2 *(22.5)* | 1.1 *( -5.7)* |

**4.3 Comparison of IPF and MDF distributions**

The GEV distribution appeared to be a generally suitable distribution for the stations in the dataset, as shown in Fig. 7. The Cramer-von-Mises test certifies a good fit for both the IPF and MDF samples, as well as for the slope and p/V corrected samples. Only a few stations lie close to the 5% line which would suggest a rejection of the null hypothesis.





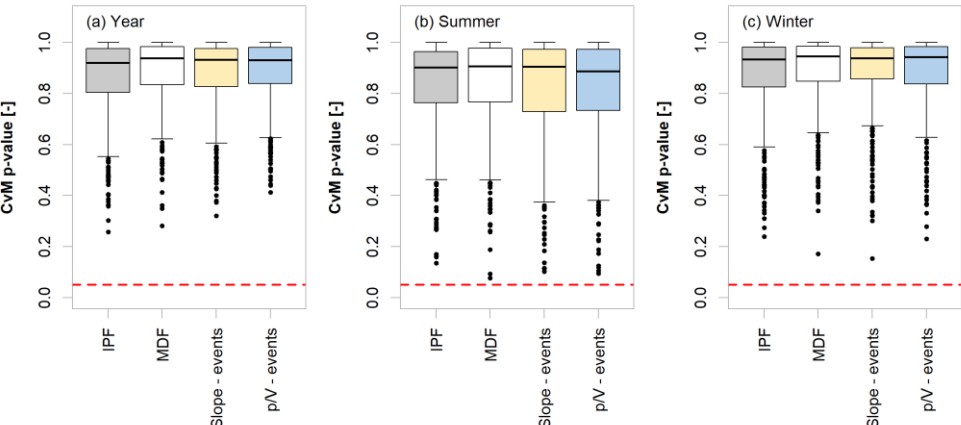

**Figure 7: Cramer-von-Mises p-values for the GEV distribution fitted to various samples of annual and seasonal maxima.**

A comparison between the estimated parameters for the IPF and MDF samples for the year and the seasons are shown in Fig. 8. As expected, the location parameters are consistently underestimated by the MDF series, with the largest errors in summer. This naturally leads to an overall downward shift of the "true" distribution when estimated from MDF values. The scales, here normalized by the location, appear to be primarily overestimated in summer, leading to distributions that are steeper for MDF than for IPF samples. For the year and winter, the error in in the scale parameter appears to be balanced in its direction.

The shape parameters differ quite substantially between the seasons. In summer the vast majority of estimated parameter values is negative, both in IPF and MDF. This indicates a heavy tail behavior for the summer floods. The fact that these negative values are in most cases significantly smaller in the MDF than in the IPF sample, suggests that the tails are overstated in the former case. This in combination with the underestimation of the location parameter leads to an overall underestimation of the lower and an overestimation of the higher flood quantiles by the MDF sample. For the year and winter, again, no clear trend

is visible.

Some distinct patterns emerge from analysis of the estimated parameters: a) higher altitude catchments have small scale and shapes close to zero. This "mild" behavior is quite consistent over all seasons and the year. b) Lowest altitude catchments generally have a large scale and, especially in summer, negative shape. These catchments thus exhibit the strongest seasonal alternation with heaviest tails in summer. At the same time these catchments show overall largest deviations between IPF and

MDF, especially in the summer half year. This suggests once more that IPF estimation for these catchments is particularly difficult.





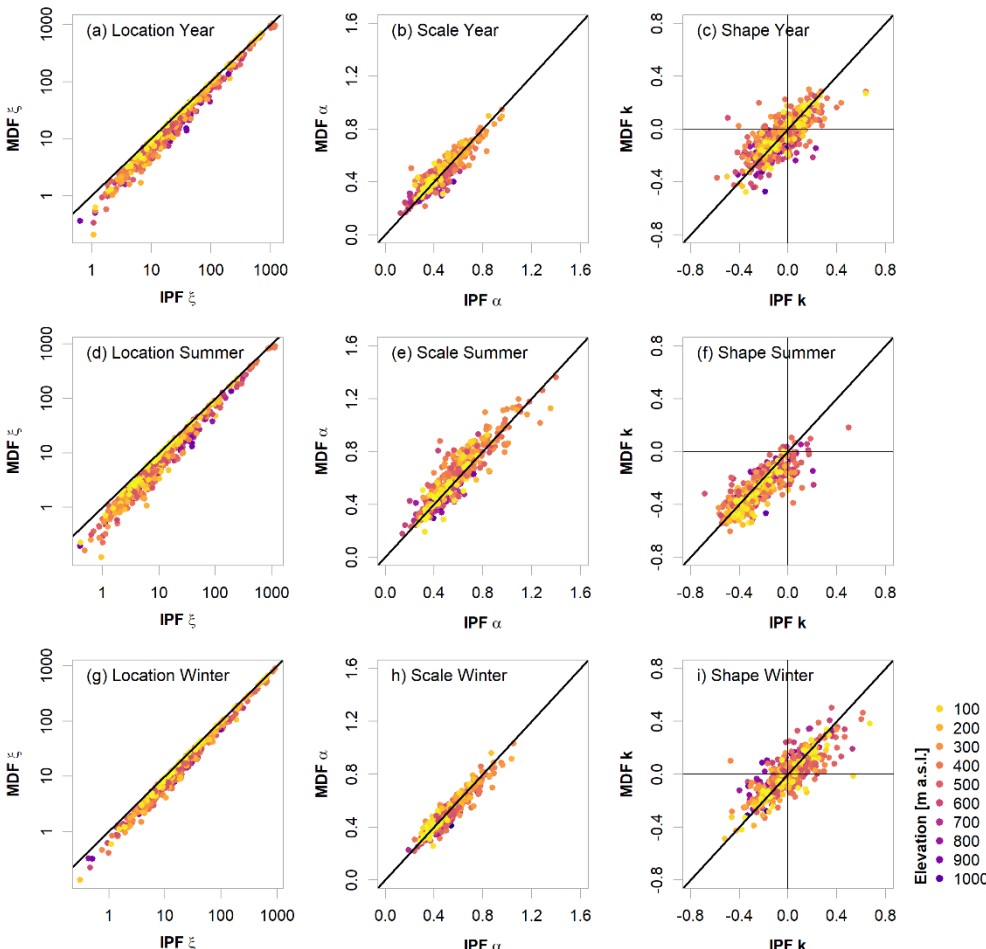

**Figure 8: Estimated GEV parameters from the IPF vs. MDF samples for the year and the two seasons.**

Generally, the heavy tails of the summer distributions in contrast to the flatter tails in winter let the summer floods become
dominant at higher quantiles. For a return period of 100 years, the summer floods exceed the winter peaks at 61.9% of the
stations. For 50 and 10 years this exceedance occurs at 51.2% and 35.7% of stations, respectively. This behavior is also
noticeable in the MDF but for fewer gauges, namely 53.4%, 43.2% and 21.0% for 100, 50 and 10-year return periods.

**4.4 Estimation of IPF quantiles**

Three approaches were tested for estimating IPF flood quantiles: a) correcting the sample L-moments required for parameter
estimation (p/V-Lmoms), b) correcting the parameters of the fitted distribution (p/V-params), and c) directly correcting the
desired flood quantiles (p/V-quant). Method a) is convenient since a single model for each L-moment facilitates a correction
of the complete distribution and hence each desired flood quantile. Estimating the L-moments has the additional advantage of
not being restricted to a certain type of probability distribution. A proper distribution can be selected and fitted locally using





the corrected L-moments. Still, the other methods may prove more robust and are hence tested as well. Additionally,
distributions were fitted to the annual and seasonal maxima that have been previously corrected using the slope (slope-events)
and p/V methods for events (p/V-events).

Fig. 9 shows the errors in parameter estimates for the different approaches in comparison to the original uncorrected MDF
error at the 490 validation stations with minimum 30-year flow records. All methods clearly improve the estimation for the
location and shape parameters, where the L-Moment correction shows overall smallest error and bias. The improvement in the
shape parameter for any of the methods is not as obvious. Since the shape parameter is generally difficult to model and the
overall error is comparably low, the direct use of the MDF shape parameter estimates is sensible. Nonetheless, the results
presented here base on estimated shape parameters.

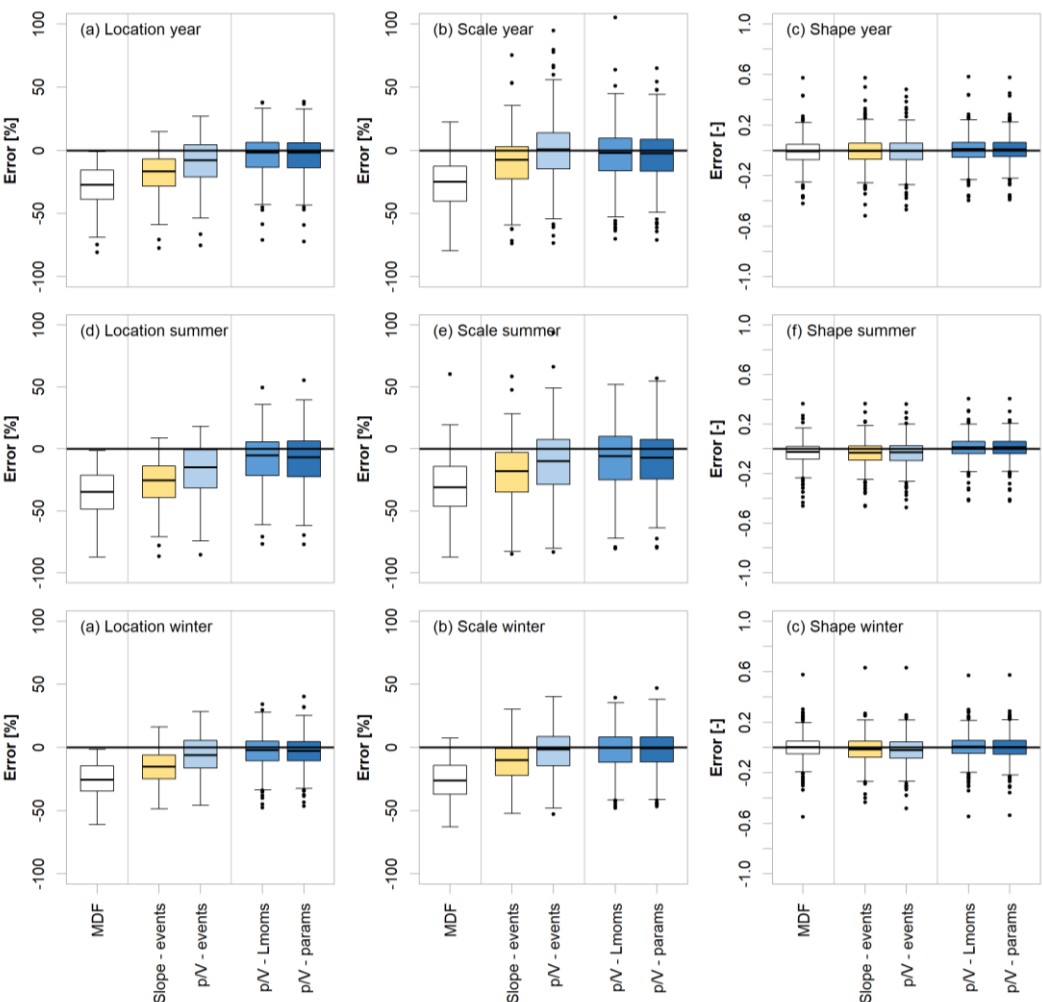

**Figure 9: Comparison of performances of various IPF-estimation methods for the GEV distribution parameters for the year and**
**the two seasons.**





Fig. 10 demonstrates the quality of the different correction approaches by means of the 10-, 50- and 100-year flood at the 490 validation stations. Other than for the mean, the differences between the individual methods are not as distinct here. It turned out that with increasing return period, the advantage of the L-Moment, parameter and quantile correction methods declines. The errors in the higher quantiles appear more random and do not relate as much to the average daily peak-volume relationship.

Moreover, even if the parameter estimates of the correction methods are generally good, slightest deviations manifest themselves in the tails of the distribution. This turns out to be especially valid for the low-altitude catchments. The overcorrection that was observed for the mean is even more pronounced here, which leads to an average decline in model performance. Last but not least, the general uncertainty in parameter estimation and extrapolation far beyond the time series length need to be kept in mind. Overall, even the estimation of the "true" IPF quantiles is potentially defective in itself, as will

be discussed in the next section.

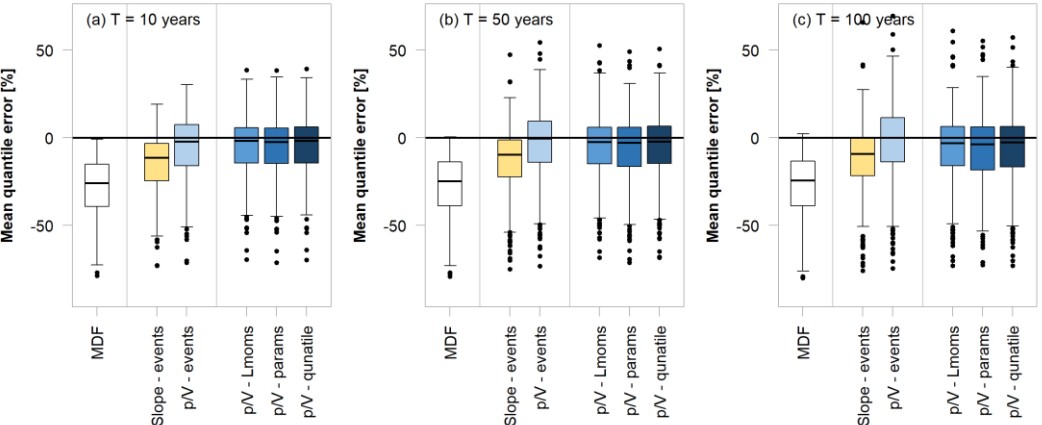

**Figure 10: Comparison of performances of various IPF-estimation methods for different flood quantiles for the entire year.**

Still, even for the 100-year flood, the L-moment and quantile approaches perform slightly better than the event correction methods, which becomes obvious from Table 3, where the normalized root mean square error (NRMSE) and the percentage

bias (PBIAS) for all methods and return periods are summarized. It appears that all three average p/V methods perform similarly well. However, due to its previously named advantages, the L-moment method is considered the superior approach in this setting.

Between the event correction techniques, the slope method performs better than the p/V in terms of overall error but is strongly biased. The outlier problem, observed for the MHQ appears to propagate severely in the p/V-event method, leading to the high

NRMSE values. When focusing on the catchments with areas below 500 km², the superiority of the p/V-methods becomes once again apparent. Still, the difference in performance between the event methods and the p/V$_{mean}$ methods decreases with increasing return period. For 50 and 100 years, the p/V-event model poses the best approach in terms of NRMSE and PBIAS, which suggests that the average p/V ratio is no longer able to properly explain the IPF-MDF relationship in the tails of the



distributions and that the severe overestimations of the event correction occur primarily at the larger catchments with overall
small deviation between MDF and IPF.

**Table 3: Performances of different IPF estimation methods in terms of NRMSE and percentage bias for different flood quantiles. The values in parentheses show the performances for catchment sizes below 500 km².**

| | T = 10 years | | T = 50 years | | T = 100 years | |
|---|---|---|---|---|---|---|
| | NRMSE [%] | PBIAS [%] | NRMSE [%] | PBIAS [%] | NRMSE [%] | PBIAS [%] |
| MDF | 17.8 *(50.0)* | -18.0 *(-32.9)* | 17.8 *(48.1)* | -18.3 *(-32.4)* | 17.9 *(47.6)* | -18.4 *(-32.2)* |
| Slope-events | 8.4 *(30.9)* | -6.0 *(-18.1)* | 10.8 *(28.6)* | -4.6 *(-16.4)* | 12.8 *(29.1)* | -4.2 *(-16.0)* |
| p/V-events | 14.7 *(23.6)* | 5.0 *( -9.7)* | 17.0 *(21.7)* | 6.3 *( -7.4)* | 18.3 *(23.0)* | 6.7 *( -6.7)* |
| p/V-Lmoms | 8.8 *(23.1)* | 1.1 *( -8.7)* | 10.0 *(24.0)* | 0.1 *( -9.4)* | 10.9 *(25.7)* | -0.4 *( -9.8)* |
| p/V-params | 8.5 *(24.2)* | 0.6 *( -9.3)* | 9.4 *(25.8)* | -0.5 *(-10.6)* | 10.1 *(27.7)* | -1.1 *(-11.4)* |
| p/V-quants | 8.9 *(23.0)* | 1.2 *( -8.5)* | 10.3 *(23.8)* | 0.6 *( -9.2)* | 11.3 *(25.5)* | 0.2 *( -9.7)* |

Finally, the model performances of the mixed models, combining summer and winter floods, are analyzed for different flood
quantiles. Their behavior is generally comparable to the annual maximum series approach, as shown in Fig. 11. Even though
the quantiles obtained with the mixed models may be more extreme and more parameters need to be estimated and corrected,
there is no indication that the IPF correction will not function in this case.

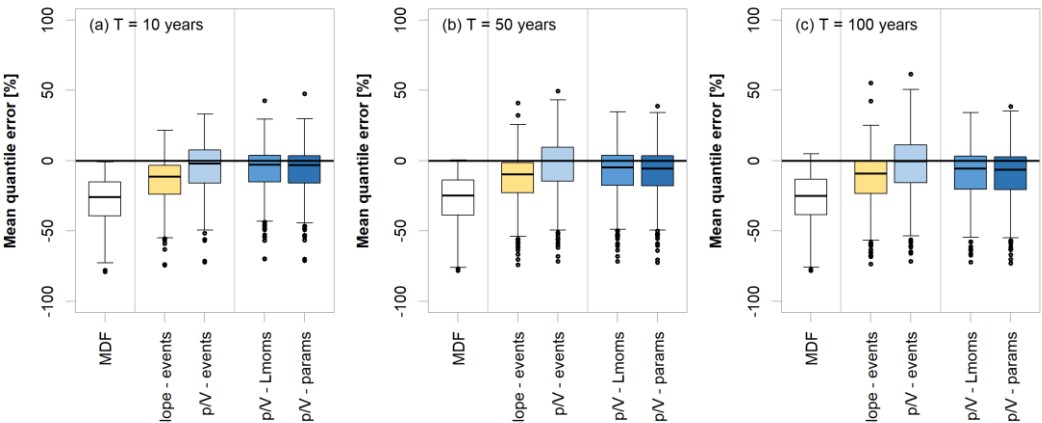

**Figure 11: Comparison of performances of various IPF-estimation methods for different seasonally mixed flood quantiles.**

The NRMSE and PBIAS values for the mixed approach are shown in table 4. Again, for the smaller catchments, the p/V-event
correction method shows the best performance.






**Table 4: Mixed-model performances of different IPF estimation methods in terms of NRMSE and percentage bias for different flood quantiles. The values in parentheses show the performance for catchment sizes below 500 km².**

|  | T = 10 years | | T = 50 years | | T = 100 years | |
|---|---|---|---|---|---|---|
|  | NRMSE [%] | PBIAS [%] | NRMSE [%] | PBIAS [%] | NRMSE [%] | PBIAS [%] |
| MDF | 17.7 *(50.2)* | -17.9 *(-33.0)* | 17.5 *(48.4)* | -18.0 *(-32.3)* | 17.6 *(48.0)* | -18.1 *(-32.1)* |
| Slope-events | 8.0 *(31.2)* | -6.1 *(-18.3)* | 9.4 *(28.4)* | -4.4 *(-16.2)* | 10.6 *(28.4)* | -3.7 *(-15.5)* |
| p/V-events | 14.6 *(23.8)* | 4.8 *( -9.8)* | 16.7 *(21.8)* | 6.4 *( -7.2)* | 17.8 *(23.3)* | 6.9 *( -6.2)* |
| p/V-Lmoms | 9.1 *(25.7)* | 0.0 *(-10.0)* | 8.6 *(26.0)* | -2.0 *(-11.4)* | 9.2 *(27.3)* | -3.1 *(-12.5)* |
| p/V-params | 9.3 *(26.1)* | -0.8 *(-10.1)* | 8.8 *(26.3)* | -2.8 *(-11.7)* | 9.4 *(27.7)* | -4.0 *(-12.8)* |

The final models for the L-moment correction can be found in Table 5. Some models exhibited a non-constant error variance

and were thus re-fitted using generalized least squares from the "nlme" R-package (Pinhero & Bates, 2018). Where this was

the case, the utilized variance function is given in the table.

**Table 5: Linear models fitted for correction of L-moments with variance function used for variance stabilization in generalized least squares fitting.**

|  |  | Model | Variance Function |
|---|---|---|---|
| L1 | Year | $L1_{MDF}$ * (1.10 - 1.27 * p/V$_{mean}$) | - |
|  | Summer | $L1_{MDF}$ * (1.28 - 1.32 * p/V$_{mean}$ - 0.00014 * elevation) | Exponential |
|  | Winter | $L1_{MDF}$ * (1.29 - 1.43 * p/V$_{mean}$ - 0.00021 * elevation) | Exponential |
| L2 | Year | $L2_{MDF}$ * (1.06 - 1.06 * p/V$_{mean}$) | - |
|  | Summer | $L2_{MDF}$ * (1.24 - 1.09 * p/V$_{mean}$ - 0.00013 * elevation) | Exponential |
|  | Winter | $L2_{MDF}$ * (1.28 - 1.37 * p/V$_{mean}$ - 0.00026 * elevation) | Exponential |
| T3 | Year | 0.95 * $T3_{MDF}$ - 0.00062 * p/V$_{mean}$ | - |
|  | Summer | 1.07 * $T3_{MDF}$ - 0.19 * p/V$_{mean}$ | - |
|  | Winter | 0.94 * $T3_{MDF}$ - 0.024 * p/V$_{mean}$ | Power |


## 4.5 Uncertainty

The results of the bootstrapping procedure used to assess uncertainty are exemplary shown in Fig. 12 for the HQ100 at a single

station with a reduced number of 100 permutations. In panel a, the IPF and MDF estimates for each permutation of the annual

maximum series are plotted against each other. This shows the bandwidths of both the IPF and MDF estimates as a result of

uncertainty in the distribution fitting. Fig. 12 b shows the estimated IPF flood quantiles vs. the quantiles estimated using the

p/V models for each permutation. The dark blue points represent the full linear models using all available stations in the study

area, while the light blue points represent 100 resampled model estimates. In this example, it becomes obvious that the range

in flood quantile estimates due to permutation in the linear models is significantly smaller than the range in estimates due to

distribution fitting. This is valid for the majority of stations.





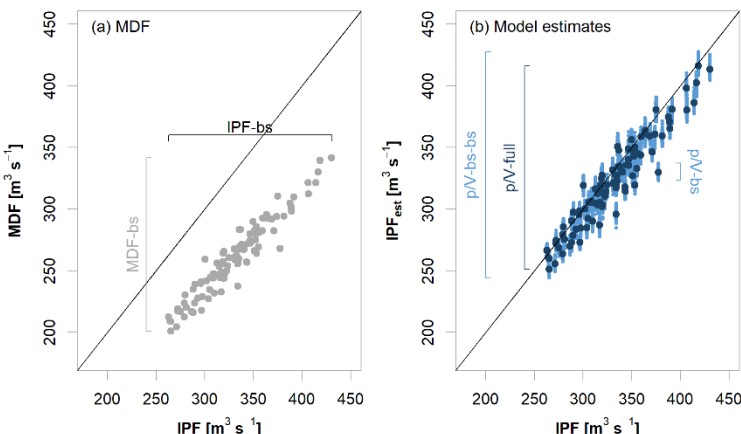

**Figure 12: Example of bootstrapping results at a station with 100 permutations. (a) HQ100 from IFP vs. MDF for each permutation of the time series, (b) HQ100 from IPF vs. 100 model estimates per permutation; the dark blue dots represent the full model.**

Fig. 13 shows the relative widths of the 95% confidence intervals for all bootstrapping samples. The average widths of the IPF-bs, MDF-bs$_s$ and p/V-full seem to be similar with a larger variability in the IPF sample. The width of the average range of the individual model permutations (p/V-bs-mean) is very small at all stations and therefore contributes little to the overall level of uncertainty (p/V-bs-bs).

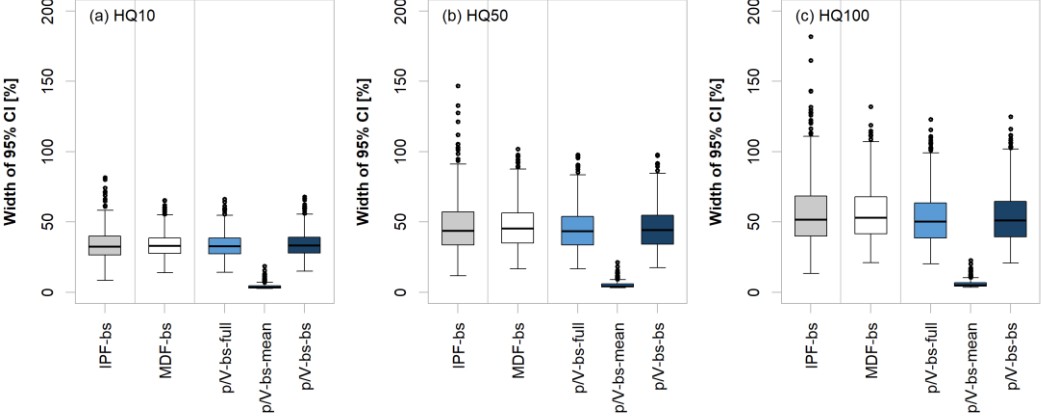

**Figure 13: Relative widths of various bootstrap samples for different flood quantiles.**

In order to assess the full bandwidth of the errors in the linear model estimates, they are compared to the range of errors in the MDF estimates. Fig. 14 shows the mean deviations from the perturbed IPF quantiles, as well as the lower and upper limits of the 50% and 95% confidence intervals of the errors for the 10-, 50- and 100-year flood quantiles. It is obvious that the overall uncertainty gets larger with increasing return period, as can be seen by the increasing distance between lower and upper confidence limits. The p/V-model estimates appear to be slightly positively biased and positively skewed, which is especially noticeable in the 95% confidence interval for the HQ100. At many stations there is a significant overestimation of the true IPF

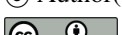



quantile with some of the linear model transpositions. The MDF estimates on the other hand exhibit the expected persistent

underestimation.

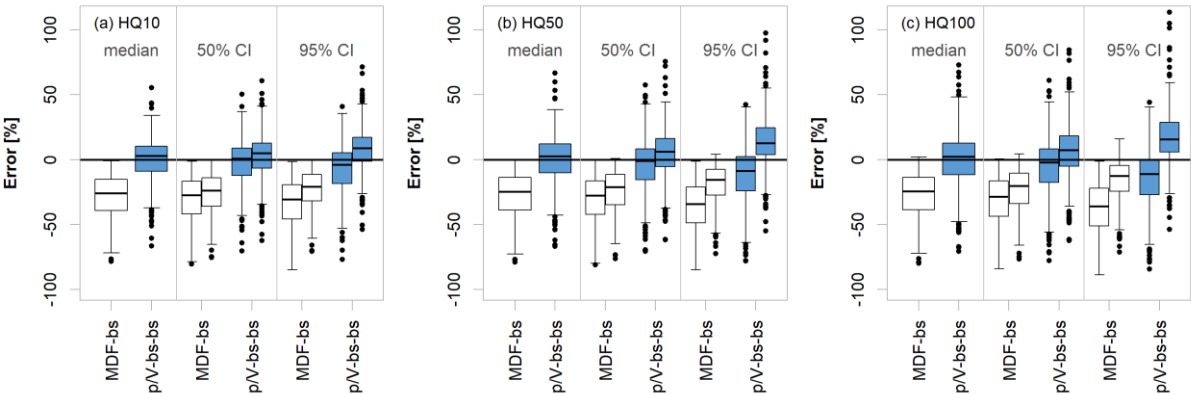

**Figure 14: Error distribution of the MDF and p/V bootstrap samples for three flood quantiles. Shown are the median errors (left), as well as the lower and upper limits of the 50% (center) and 95% confidence intervals (right).**

Fig. 15 summarizes the general overlap of the confidence intervals of MDF and estimated IPF with the confidence intervals of

the observed IPF for the three flood quantiles. It becomes obvious that the agreement between IPF and the p/V model estimates

is significantly larger than with the MDF values. This observation suggests that with high probability the p/V model estimates

are in the range of the "true" IPF quantiles.

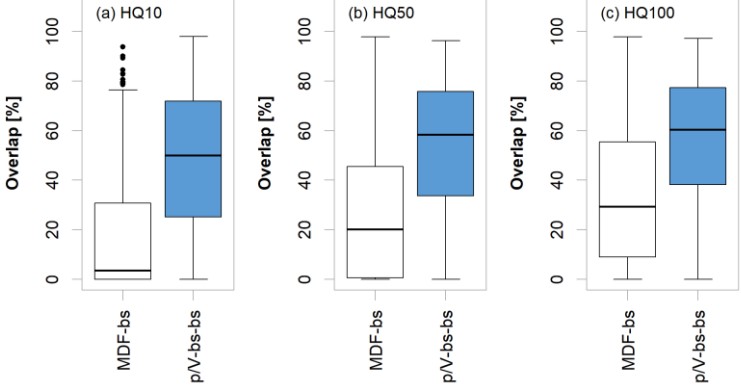

**Figure 15: Percentage overlap between the 95% confidence intervals of all IPF bootstrap estimates and MDF and p/V model bootstrap estimates for three flood quantiles.**

## 4.6 Range of applications and limitations

The method of correcting the error of MDF floods via p/Vs performs well and is easily applicable in our study area. However,

its great simplification and mere approximation of physical flood generating processes results in some problems and limitations

that will be listed and discussed here.





The first aspect that may influence the performance of the proposed IPF correction method is the event separation technique. The chosen technique determines how flood events and thus the required hydrograph characteristics are defined. The choice of baseflow separating algorithm can greatly affect the identification of start and end points of flood events. Strict independence criteria and thresholds for event recognition may lead to rejection of crucial flood events when considering daily time series.

Lax criteria, on the other hand, may create unnaturally long multi-peak events and false inclusion of small events, both leading to unrealistic hydrograph characteristics and IPF estimates. Thus, the additional step of refining multiple peak events, as suggested by Tarasova et al. (2018) should be carried out, when rainfall and snowmelt information is available.

Using the $p/V_{event}$ in order to correct individual events and then using the corrected series for FFA poses in theory a more sensible approach than using the $p/V_{mean}$ from the annual MDF maxima. As mentioned before, maximum MDF events do not

necessarily coincide with maximum IPF events, which is why correcting all events first and then selecting the annual maxima should yield a more appropriate IPF sample. But again, correcting individual events depends greatly on a very careful event separation, which could not be achieved in this case and which led to some unrealistic IPF estimates. Nonetheless, if a proper event separation is possible, the event correction method may have the larger potential. In such a case, a single model would be sufficient to account for all aspects of IPF estimation, including high flood quantiles.

A problem for IPF correction, which has been exhaustively discussed above, are gauges that exhibit little difference between MDF and IPF floods, even though their p/V ratio would suggest a much larger error. For our dataset this applies to the lowest-altitude gauges in the dataset. The MDFs at these stations are overcorrected and thus exhibit severe overestimation of the true IPFs. We therefore discourage the application of the suggested correction methods at gauges that are both situated below 100 m a.s.l. and have catchment areas larger than 200 km².

This observation may also suggest that other factors need to be considered for proper error estimation or that the parameters of the correction models need to be adjusted for different subsets of data. This is also relevant for the question of universality of the proposed method. Our data set is limited and representative of a temperate humid climate and moderate altitude. Thus, a qualitative sensitivity analysis is carried out on the full 653-stations dataset in order to identify patterns that may be extrapolatable to other regions. The subsets are selected by combinations of geographical location, catchment size and gauge

elevation. Target variable is the mean annual maximum IPF. Differences in the individual models due to different degrees of freedom are natural, which is why only those subsets that lead to significant deviations from the original model are mentioned here.

Two sets of stations deviate noticably from the original model. The first one includes the low-altitude gauges discussed before. Here the overall error is so small that no correction yields better results than correction by the linear model. The second group

includes the catchments with areas below 50 km². The errors for these stations appear very scattered and randomly distributed. Comparing the p/V from the daily series with the p/V obtained from instantaneous events, it becomes obvious that the difference increases with decreasing catchment size and becomes excessively large and random for catchment sizes below 100 km². The correction using mean daily p/V only functions where unknown instantaneous flood dynamics are roughly approximated by observed daily flow variability. The smaller the temporal scale of an instantaneous flood event, the poorer it



is reproduced in the daily records. If instantaneous events manifest themselves primarily on a subdaily basis, the possibility to describe their dynamics via daily flows becomes ineligible. This observation is also in accordance with the observed temporal shifts between MDF and IPF events, which is increasingly pronounced in smaller catchments. In summary, the proposed correction method founders at smaller scales below 100 km². Even though the IPF estimation leads to a general improvement at this scale, the daily flood time scale poses a poor predictor in these catchments.

Longitude and latitude do not appear to have any effect on the model fitting. Dividing the study area into quadrants does not result in any differences between the subsets, even when equalizing the other factors catchment size and elevation. Also, neither record length nor period of record appear to have an influence.

The distinction between summer and winter for representation of the two most opposite flood types is particularly valid for this study area and should be adjusted where flood types are otherwise distributed. In general, even the rough distinction

between different flood types for IPF estimation proved meaningful in our case, as it revealed different dynamics and MDF-IPF relationships. This observation could be further exploited by more carefully defining and distinguishing flood types, as e.g. proposed by Fischer (2018) or Tarasova et al. (2020).

Finally, one should note that the type of distribution for flood quantile estimation can only be selected based on daily data and may differ from the optimal IPF distribution. For our data, the GEV proved flexible enough to be a good match in both MDF

and IPF but this could differ in other cases.

## 5 Conclusions and Outlook

As in other studies before, it could be shown that the IPF-MDF relationship depends primarily on catchment size. It could also be observed that other factors, in this case gauge elevation, play a role in determining the difference between MDF and IPF floods. The relationship also appeared to differ between the two types of floods considered here, namely winter and summer

floods. Since summer floods are often caused by short but intense rain events and thus exhibit steep rising and falling limbs, their subdaily peaks are much larger than and difficult to estimate from the smoothed average daily peaks. Long, voluminous winter floods on the other hand show a much smaller IPF-MDF ratio and are easier to model.

This study has also shown that hydrograph characteristics, like the peak-volume ratio of flood events can be used to estimate instantaneous peak flows when only average daily series are available. The p/V ratio may be used to predict both IPFs of

individual events and instantaneous flood statistics, including mean annual and seasonal maximum flows and flood quantiles. Due to improper flood event separation, the event-correction method produced some outliers in our case but may work significantly better when flood events can be defined more carefully. In general, the p/V method requires a minimum of data and can be applied using mere information from the daily series itself. The performance could be marginally improved by including gauge elevation as additional predictor in the models.

The general recommendation for estimating IPF flood quantiles is to use the average p/V approach for correction of L-moments. This method is convenient since L-moments can be globally corrected while distributions may be locally fitted



afterwards. It turned out that the first two L-moments are easily estimated using $p/V_{mean}$, while higher order L-moments or L-moment ratios are more difficult to model with this approach.

There are two limitations, where the proposed method should be handled with care: a) at stations with an elevation below 100 m and catchment areas above 200 km², since it overestimates the true difference between IPF and MDF and b) at catchments smaller than 100 km², where it underestimates the error so that the full correction potential cannot be achieved. Still, in comparison to the slope method, the time-scale approach works significantly better for smaller catchment areas, especially below 500 km². For larger catchments, the two methods are more comparable in performance.

For future analyses it will be meaningful to test the universality of the proposed approach in other study regions. Also, the
effect of the flood event separation on the IPF estimation performance should be analyzed in more detail, especially in order to improve the event correction technique. Finally, it will be interesting to see if explicit consideration of more carefully defined flood types can improve the IFP estimation in mixed models.

**Data availability**

The discharge data used in this study is publicly available on the websites of the respective federal agencies.

Lower Saxony: Niedersächsischer Landesbetrieb für Wasserwirtschaft, Küsten- und Naturschutz (NLWKN) http://www.wasserdaten.niedersachsen.de/cadenza/

Saxony-Anhalt: Landesbetrieb für Hochwasserschutz und Wasserwirtschaft Sachsen-Anhalt (LHW) https://gld-sa.dhi-wasy.de/GLD-Portal/

Saxony: Sächsisches Landesamt für Umwelt, Landwirtschaft und Geologie (LFULG) https://www.umwelt.sachsen.de/umwelt/infosysteme/ida/

Bavaria: Bayerisches Landesamt für Umwelt (LfU) https://www.gkd.bayern.de/de/

Baden-Württemberg: Landesanstalt für Umwelt Baden Württemberg (LUBW) https://udo.lubw.baden-wuerttemberg.de/public/


**Author contribution**

UH formulated the research goal. The study was designed by both authors and carried out by AB. AB prepared the manuscript with contributions from UH.

**Competing interests**

The authors declare that they have no conflict of interest.

**Acknowledgements**

This work is part of the research group FOR 2416 "Space-Time Dynamics of Extreme Floods (SPATE)" funded by the German
Research Foundation ("Deutsche Forschungsgemeinschaft", DFG).





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
