# Peer review of "Flood frequency analysis using mean daily flows vs. instantaneous peak flows"

_Hydrology and Earth System Sciences, 2021_

## Referee Comment (RC2)

**Overview**

This manuscript compares flood frequency analysis using mean daily flow (MDF) and instantaneous peak flow (IPF) and proposes the peak-volume ratio method (p-V method) to estimate IPFs from MDFs. A simple baseflow separation is needed before applying the p-V method. The p-V method and the slope method proposed by Chen et al (2017) are applied to ~ 450 German basins across various climate regions. The authors reported that:

    a) The discrepancy between MDF and IPF depends on catchment size and elevation;
    b) The p-V method performs well for basins with sizes ranging from 100 to 200 $km^2$;
    c) The p-V method can be applied both at the event and annual (MHQ) scales.

**General comments**

This work aims to promote the flood frequency analysis technique, which is interesting and attractive. The thinking of using peak-volume ratio of direct runoff hydrographs to quantify hydrograph shapes and ultimately estimate IPFs from MDFs is creative. The study is based on the investigation of ~450 catchments with rich flood generating mechanisms. The story flow of the manuscript is well. Few concerns need to be addressed prior to the publication. Detailed comments to the authors, which might be helpful to improve this manuscript are given below.

1. The authors stated that except for catchment size, gauge elevation (I assume average elevation of each basin) plays a role in determining the differences between MDF and IPF (see lines 9-10, 458-459, 468-469). It would be helpful if the authors provide more quantitative evidences that adding the gauge elevation (I assume it is the average elevation of each basin) predicting factor is beneficial, for example, comparing the performances between models with the elevation predictor switching on and off.
2. Figure 2: The design of the figure is nice and I think it would be good to provide a quantitative descript of the catchments studied, for example, add a table or a figure to report the number of catchments in each range of sizes and altitudes. It would be even better to provide the average prediction errors of the p-V method for each range of basins because this information might be helpful for potential users.

**Specific comments:**

1. line 63 equation 1 and Figure 2: Maybe it is more appropriate to use the term "discrepancy" rather than "Error"?
2. Figure 1: Please present the types of stations in the legend in English.
3. Line 224 and Table 1: it would be helpful to provide the dependent variable, statistical significance, and coefficient of determination of these linear models.
4. Line 279: delete the duplicate "in".
5. Lines 333-335: Outliers could be part of the explanation to why NRMSE for p-V method is larger. Many of the outliers for the p-V methods are concentrated in basins

larger than 500 km$^2$, and errors for these basins are often positive (Figure 5 c,d), indicating that the p-V method appears to overestimate IPFs for large basins (at least in comparison to the slope method, and apparently the slope method tend to underestimate IPFs for small basins). It would be nice to clearly present this point at the result section.

6. Line 478: This last sentence needs evidence. According to Figure 5, tables 2,3,4, it appears that the slope method does not overpredict IPFs for larger basins.